# *Dothistroma septosporum* Not Detected in *Pinus sylvestris* Seed Trees from Investigated Stands in Southern Poland

Pola Wartalska [1], Tomasz Oszako [1,2], Sławomir Bakier [2], Lassaâd Belbahri [3], Tadeusz Malewski [4], Tom Hsiang [5], Elżbieta Popowska-Nowak [6] and Justyna Nowakowska [6,*]

1 Department of Forest Protection, Forest Research Institute, Braci Leśnej 3, 05-090 Sękocin Stary, Poland; p.wartalska@gmail.com (P.W.); T.Oszako@ibles.waw.pl (T.O.)
2 Institute of Forest Sciences, Faculty of Civil Engineering and Environmental Sciences, Bialystok University of Technology, Wiejska 45E, 15-351 Bialystok, Poland; s.bakier@pb.edu.pl
3 Laboratory of Soil Biology, University of Neuchâtel, 2000 Neuchatel, Switzerland; lassaad.belbahri@unine.ch
4 Museum and Institute of Zoology Polish Academy of Science, Wilcza 64, 00-679 Warsaw, Poland; tmalewski@miiz.waw.pl
5 Environmental Sciences, University of Guelph, Guelph, ON N1G 2W1, Canada; thsiang@uoguelph.ca
6 Institute of Biological Sciences, Faculty of Biology and Environmental Sciences, Cardinal Stefan Wyszynski University in Warsaw, Wóycickiego 1/3 Street, 01-938 Warsaw, Poland; e.popowska-nowak@uksw.edu.pl
* Correspondence: j.nowakowska@uksw.edu.pl; Tel.: +48-22-569-68-38

**Abstract:** In recent years, the decline of pine stands in Europe, including Poland, has been caused by the emerging needle pathogen *Dothistroma septosporum*. Although this fungus appears to preferentially infect *Pinus pini*, *P. pinaster* or *P. radiata* in Southern Europe, it has been reported in stands of *P. nigra*, *P. mugo* and *P. sylvestris* from Southern Poland. Our preliminary tests of symptomatic needles of diseased pines, including black pine (*P. nigra*), showed the presence of both *D. septosporum* and *D. pini*—the latter as the first report in Poland. No other endophytic pathogen, i.e., *Lecanosticta acicola* or *Cenangium ferruginosum*, were found. More extensive molecular surveying based on β-*tub2* amplification of DNA in needle samples from 72 seed trees of *P. sylvestris* in nine different Forest Districts of Southern Poland did not find the presence of *D. septosporum*. Our study revealed that the seed trees from which we collected propagation material were free from the pathogen, and its endophytic behavior was not confirmed in our testing. Consequently, these investigated trees of *P. sylvestris* should be suitable for seed collection and propagation, following the requirements of "good" phytosanitary quality as "pathogen-free" pine seeds used for reforestation.

**Keywords:** *Dothistroma pini*; dothistroma needle blight (DNB); forest reproductive material (FRM); molecular analysis; Scots pine

## 1. Introduction

Pine trees are an important component of native forests and plantations in Europe, where they have an economic importance and play an important ecological role. Alien invasive pathogens of conifers such as *Lecanosticta acicola* (von Thümen) Schischkina & Tsanava, *Sphaeropsis sapinea* (Fr.) Kyko & Sutton, *Dothistroma* spp., or *Fusarium circinatum* Nirenberg & O'Donnell have recently caused the death of many pine trees in Europe [1,2]. Some oomycetes, e.g., *Phytophthora cinnamomi* Rands, or fungi, e.g., *F. circinatum*, now found only in southern Europe (mainly in Spain and Portugal), can easily spread to new geographical locations as a result of global warming, posing the risk of further northward spread [3]. Some studies have shown that *F. circinatum* may also cause a large amount of damage in Poland if introduced there [4].

Dothistroma needle blight (DNB) is one of the most important diseases of pine and has a global distribution [5,6]. More than 80 pine species, including *Pinus nigra* (Arn.), *P. brutia* (Ten.), *P. pinaster* (Ait.), *P. pinea* (L.), *P. mugo* (Turra), and *P. sylvestris* (L.) are

known to be susceptible to this pathogen [7,8]. Under favorable conditions with high infection pressure, infections have been reported in other tree species, such as *Larix decidua* (Mill.), *Pseudotsuga menziesii* (Mirb. Franco), *Picea abies* (L. Karst.), *Picea omorika* (Pančić Purk.), and *Picea sitchensis* (Bong. Carrière) [6,7]. Based on morphological and molecular studies [5], the pathogen formerly known as *Mycosphaerella pini* [9] has been subdivided into two distinctly different species: *Dothistroma septosporum* [10] (Dorog.) M. Morelet (*Mycosphaerella pini* E. Rostrup) [11] and *Dothistroma pini* (Hulbary). These fungi belong to the phylum Ascomycota, in the family Mycosphaerellaceae. They are mainly known as pine needle damaging factors, producing dothisthromin, a toxin responsible for red discoloration of lesions on the needles of infected plants [5].

*Dothistroma septosporum* is a cosmopolitan pathogen first found in Russia in 1910, and was at that time called *Cytosporina septospora* (Dorogin) [10]. Connections between remote populations of *D. septosporum* in north-east Europe and Asia were detected by cluster analyses, and similar genetic patterns were found in Norway, Serbia, and the Russian Far East. Large-scale effects of the disease were not observed until 1950–1960 in the Southern Hemisphere [12], and now it has been reported in more than 44 different countries [6]. In many countries, it has caused severe damage, especially in forest nurseries. Chile, New Zealand, and Kenya, where plantations consist mainly of monocultures of introduced susceptible hosts, have suffered enormous economic losses due to the pathogen [13]. In the past 20 years, the disease has also become a serious problem in the Northern Hemisphere, reducing pine production and ravaging native pine forests [12,14–16]. The occurrence of DNB has been associated with the Southern Hemisphere and the planting of exotic trees susceptible to this pathogen, such as *Pinus radiata* D. Don, *P. ponderosa* Douglas ex C. Lawson, and *Pinus nigra* J. F. Arnold [6].

In Europe, DNB was first reported in 1907 in France [11,17,18], and then in 1954 in England [19], 1955 in Serbia [20], 1969 in Greece [21], 1971 in Slovenia [22], 1974 in Spain [23], 1976 in Italy [24], 1983 in Germany [25], 1990 in Hungary [26], 1996 in Slovakia [27], 2000 in Czech Republic, 2002 in Lithuania [28], 2006 in Estonia [29], 2007 in Finland and Sweden [30], 2008 in Latvia and Russia [31,32], 2009 in Norway [33], and more recently in the Ukraine, Belarus, and Switzerland [30]. A detailed chronology of the appearance of *Dothistroma* spp. in Europe was provided by Drenkhan et al. [6].

In Poland, DNB was first reported in 1990 on *P. nigra* in Domiarki in the southwestern part of the country, and was observed regularly thereafter [34]. Until 1998, *D. septosporum* in Poland was only reported on *P. nigra*, but not on *P. sylvestris* or other conifer species [34]. However, a study conducted in 2014 [35] found the presence of *D. septosporum* at 37 new sites of black pine stands, mainly in southern Poland, although the presence of this pathogen was also detected in a few *P. sylvestris* and *P. mugo* Forest District (FD) locations, i.e., Brynek (Świerklaniec FD), Bzowo (Dąbrowa FD), Czernichów (ornamental nursery), Dębowiec (Prudnik FD), Domiarki (Miechów FD), Kamyk (Krzeszowice FD), and Leśnice (Lębork FD) [35]. Therefore, our research focused on seed trees of *P. sylvestris*, growing in areas adjacent to commercial forests where DNB has been observed. To date, *D. pini* has not been recorded in Poland.

The macroscopic symptoms and morphological characteristics of *D. septosporum* and *D. pini* are easy to confuse due to their close similarity. Recent research on *Dothistroma* has focused on the detailed studies of the genetic structure of Mycosphaerellaceae pathogens, which has helped to confirm the region of origin and pathways of the spread of DNB globally [36]. Additionally, a PCR protocol was developed by Ioos et al. [37] for rapid detection of *D. septosporum* and *D. pini*, in addition to *Lecanosticta acicola* [(von Thümen) Schischkina & Tsanava] (the causal agent of brown spot disease), with which the two *Dothistroma* species are often confused [1]. In addition to *L. acicola* causing new emerging diseases in many pine species in Europe, USA, and Canada, *Cenangium ferruginosum* (Fr.) leads to shedding of pine needles, and in consequence, the death of the tree [1,38,39]. Molecular methods are used in a variety of ways, including to identify and distinguish fungal species, to detect and identify species from plant samples taken from herbaria, and

to identify the pathogen from collections of plant material associated with older literature that did not use these more modern techniques [17].

The chain of pine stands stretches across Europe from the Iberian Peninsula through Central Europe to Russia. In Central Europe, particularly Poland, pine trees occur in over 58% of the state forests [40], and they are prone to numerous pathogenic infections [41]. The global distribution of *D. septosporum* means that these fungi have readily been transferred to new environments, most likely via seeds [8]. We assumed that *D. septosporum* can spread from needles to seeds and onward to forest nurseries. Some plant endophytes are known to be transferred by a process known as vertical transfer under the seed coat [42]. In contrast to horizontal transfer, i.e., direct transfer of an endophyte from parent to offspring, transfer under the seed coat provides a ready source of inoculum for germinating seeds [43,44]. Although it has not been confirmed that pathogens such as *D. septosporum* can be transmitted under the seed coat, it is known that spores of pathogenic fungi can be transferred to pollen [45,46]. Some fungi, such as *Cyclaneusma minus* (Butin) DiCosmo, Peredo & Minter, a pathogen of needles, are able to infect and behave as endophytes throughout their life cycle [47]. Furthermore, latent infections in *P. nigra* and *P. sylvestris* by *D. sapinea* without symptom development have also been observed [48].

Needlecast of pine trees is common in European forests. The causal pathogens include *D. septosporum*, *D. pini*, *C. ferruginosum*, and the recently introduced *L. acicola* [1,34,38,39,49]. *Dothistroma septosporum* occurs globally, whereas *D. pini* has been found in the USA and in more than ten European countries, such as France, Hungary, Ukraine, and Switzerland [17,30]. The records closest to Poland were in the Czech Republic and Ukraine [30,50], from where it may have spread to Poland. Indeed according to Drenkhan et al. [50], there was a migration of *D. septosporum* from the Czech Republic to Finland through Poland.

The legislation in force within the European Union aims to eliminate the presence of pathogens in parts of plants intended for planting, thereby reducing but not eliminating the risk of spreading pathogens within Europe. Good forestry practice requires that forest reproductive material (FRM) is free from harmful pathogens, which is in line with Recommendations of the European Commission Council Decision 2008/971/EC on FRM of the "qualified" category. Foresters use selected "seed trees" for seed production and these trees (sometimes called "mother trees" or "plus trees") are selected according to criteria that show the most desirable characteristics, such as height, diameter, or general health of the tree [51,52].

During a visual inspection of *P. mugo* planted as an ornamental shrub in front of the Forest Protection building (IBL) in Sękocin Stary near Warsaw in central Poland, symptoms were found on discolored needles with many dead shoots (Figure 1). Microscopic identification followed by DNA extraction and PCR analyses revealed the pathogen to be *D. septosporum* (unpublished data based on 12 samples). During the same period, we were asked to explain causal factors of needle disease from black pine (*P. nigra*) and Scots pine (*P. sylvestris*) planted in a private garden near Warsaw, from an ornamental nursery, and from a nearby forest stand, and this led to the current study. After hearing reports of diseased pine trees in southern Poland [35], we conducted the current research to determine whether the pathogen was present in forests containing *P. sylvestris* seed trees, from which cones have been collected and used for regeneration. Another purpose of the current study was to assess whether the pathogen may persist as an endophyte and be present in asymptomatic needles of healthy pines.

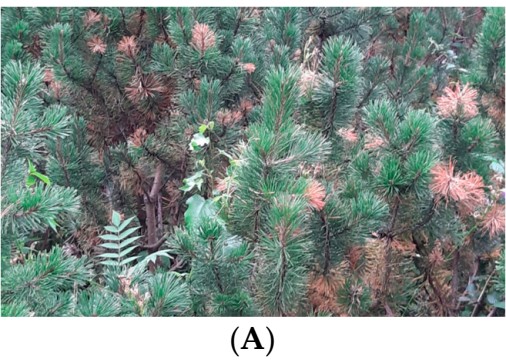

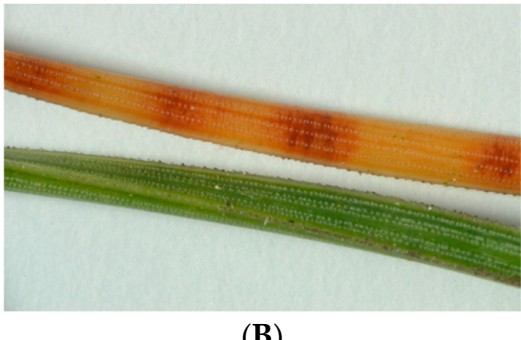

(A)    (B)

**Figure 1.** Yellowed discoloration of needles of *Pinus mugo* caused by *D. septosporum* in the greenhouse surrounding the Forest Research Institute in Sękocin Stary, Poland (**A**). Symptoms of red band pine disease: needles with brown, necrotic tissue of *D. septosporum* developing under the epidermis of *P. mugo* needle (**B**). Photos by C. Bystrowski.

## 2. Materials and Methods

### 2.1. Preliminary Assessment

The aim of this preliminary study was to determine whether target fungi (needle pathogens) were abundant in the plant material. The selection criteria for the preliminary study were broad because there was limited information about potential DNB. The criteria for the preliminary study arose from information received by the Polish IBL Forest Protection department on the disease symptoms of pine trees in different regions of the country. Therefore, informants were asked to send samples of symptomatic pine trees from which needles were collected for further laboratory analysis. In the course of carrying out consultation assessments at the request of both private and public entities, some needles were collected from five dying Scots pines (*P. sylvestris*) and four black pines (*P. nigra*) originating from Central Poland, including private gardens near Warsaw in Konstancin Jeziorna (sample No. 1–7, Figure S1) and from an ornamental nursery in Osuchów (No. 8). We also analyzed some symptomatic needles of one Scots pine from northeastern Poland (the Pomorze Forest District, Dobiesz Forestry—No. 9). All samples were subjected to the visual inspection of plant tissues, from which DNA was extracted and analyzed using PCR and specific primers for *D. septosporum*, *D. pini*, *Lecanosticta acicola*, and *Cenangium ferruginosum*, as described in Sections 2.3 and 2.4.

### 2.2. Main Study to Detect Pathogens in Seed Tree Needles

The selection criteria for the main study were carefully chosen based on the available literature [35] and the information we received from the international COST–Action FP1102 "Determining Invasiveness And Risk Of Dothistroma (DIAROD)". These showed that *D. septosporum* is present in commercial stands of both Scots pine and black pine in southern Poland. Investigated stands were selected in southern Poland, not adjacent to commercial stands where the disease had been detected, but where seed trees were available for sampling [35].

To determine whether the shoots of selected Scots pine trees were free of needle pathogens, i.e., *D. septosporum*, *D. pini*, *L. acicola*, and *C. ferruginosum*, we decided to focus only on those pathogens which were found in the preliminary study. Our working hypotheses were that the foliage of seed trees was free of the above pathogens, and that these pathogens did not survive as endophytes in needle tissues.

#### 2.2.1. Sampling of Plant Material

Collection of plant material was carried out between 2002 and 2005 in nine selected FDs in Southern Poland, i.e., Józefów, Narol, Biłgoraj, Janów Lubelski, Kraśnik, Rudnik, Krasnystaw, Niepołomice, and Tomaszów [53] (Figure 2). The forest sampling strategy involved using a firearm to shoot twigs from the highest parts of the upper crown to collect at least two asymptomatic needles from every other tree, along two or more 50 m transects.

The plant material for DNA testing (needles) was obtained from the selected trees (mother trees) randomly distributed in a given stand, at least 25 m away from each other, to avoid close kinship between the studied individuals, and appearing healthy. These seed trees had been identified by the Polish State Forest Administration as possessing exceptional characteristics. A total of 144 needles of *P. sylvestris* were collected from eight individuals in each of nine different stands (Table S1). The age of the stands ranged from 90 to 126 years and the forest types were typical of local conifer forests in Southern Poland. Three of nine populations (Niepołomice, Biłgoraj, and Janów Lubelski FD) were classified as seed reservoirs for use as seed sources, and the remaining six populations were classified as commercial [51]. Plots were selected to be representative of the whole country and not to deviate from the national average in terms of the level of genetic variation (heterozygosity and coefficient $F_{ST(WC)}$ [54]. These statistics were established from earlier sampling studies, e.g., [53]. All needles were frozen at $-80\,°C$ before further investigation.

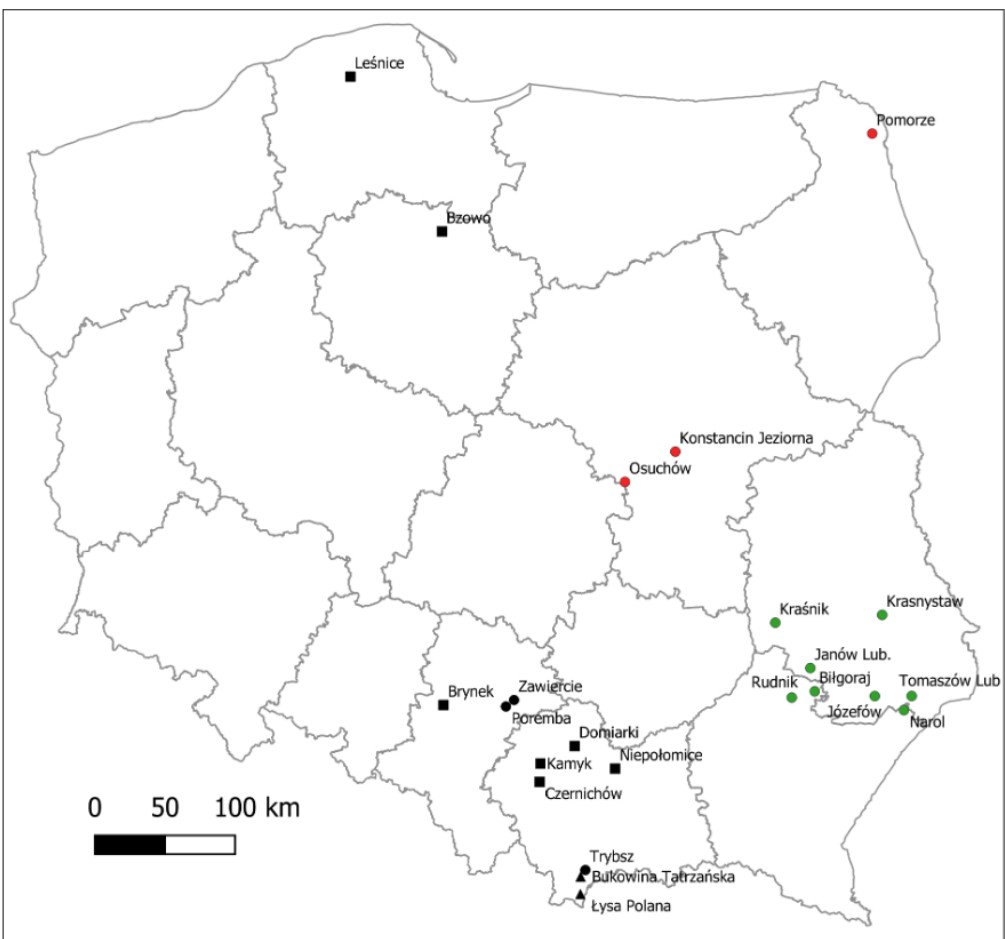

**Figure 2.** Contour map of Poland with marked locations of Forest Districts in which plant material used for *D. septosporum* testing was collected (green circles). Most of those sites were typical of coniferous forests, with a mean age of 103 years (cf. Table S1). Red circles—samples of pine needles taken for the preliminary study (Figure S1). Black circles represent collected available *Pinus sylvestris*, black triangles—*P. mugo*, and black squares—*P. nigra*, where *D. septosporum* was detected by Boroń et al. [35].

### 2.2.2. Preparation of Fungal Positive Control

Pure cultures were established only for fungal species detected on pine needle samples examined in the preliminary collection. The pure culture of *D. septosporum*, which served as the positive control in our study, was kindly provided by Prof. Piotr Boroń from the

University of Agriculture and Forestry, Faculty of Forestry, Krakow (Poland). Cultures were maintained on malt extract agar and kept at 4 °C.

### 2.3. Genomic DNA Isolation

Plant material from preliminary and main studies (PS and MS, respectively), and from *D. septosporum* pure culture (Figure 1 and Figure S1), was subjected to DNA isolation. Genomic DNA for each sample was obtained from 100 mg of symptomatic and asymptomatic needles, or 50 mg of pure culture using the NucleoSpin® Plant II kit (Macherey-Nagel GmbH & Co., Düren, Germany) according to the manufacturer's instructions. The quantity and quality of the obtained DNA was assessed spectrophotometrically at 230, 260, and 280 nm in a NanoDrop® ND-1000 (Wilmington, DE, USA).

### 2.4. Molecular Analysis of the Sampled Material

The genomic DNA samples from needles from PS and MS, and from the *D. septosporum* pure culture, were amplified using the same protocol but adjusted for primers (Table 1). Molecular testing for *Dothistroma* spp. was based on amplification of a *D. septosporum* marker gene of the beta-tubulin subunit 2 (β-*tub2*) [34,42]. Detection of *D. pini* and *L. acicula* used the set of primers for the translation elongation factor *EF1-α* [37,55], and the detection of *C. ferruginosum* was based on the internal transcribed spacer (ITS) of ribosomal DNA [56].

**Table 1.** Primers used for PCR detection of *Pinus* pathogens.

| Species | Gene | Primer Name | Primer Sequence | Expected Fragment Length (bp) | Ref. |
|---|---|---|---|---|---|
| *Dothistroma septosporum* | β-tubulin | DStub2-F DStub2-R | 5′-CGAACATGGACTGAGCAAAAC-3′ 5′-GCACGGCTCTTTCAAATGAC-3′ | 231 | [37,55] |
| *D. pini* | EF1-α | DPtef-F | 5′-ATTTTTCGCTGCTCGTCACT-3′ | 193 | [37,55] |
| | | DPtef-R | 5′-CAATGTGAGATGTTCGTCGTG-3′ | | |
| *Lecanosticta acicola* | EF1-α | LAtef-F LAtef-F | 5′-GCAAATTTTCGCCGTTTATC-3′ 5′-TGTGTTCCAAGAGTGCTTGC-3′ | 237 | [37,55] |
| *Cenangium ferruginosum* | ITS | CfF CfR | 5′-GA*TCATTAC CAGAAGTGTCC-3′ 5′-CCTAGGTGA GTTGGGGTTGC-3′ | 477 | [56] |

Polymerase Chain Reaction (PCR) was performed in a total sample volume of 20 μL, in a Veriti 96 Well AB thermocycler (Applied Biosystems, CA, USA). Each sample contained the following 3 μL of genomic DNA, 10 μL of RedTaq PCR ReadyMix (Sigma-Aldrich, Milwaukee, WI, USA), 1 μL each of the forward and reverse primers at 5 μM, 1 μL of DNA extract, and sterile water to a final volume of 20 μL. Thermocycling conditions for detection of *D. septosporum*, *D. pini*, and *L. acicola* used targeted primer sets (Table 1) consisting of 10 min initial denaturation at 95 °C followed by 35 cycles of denaturation at 95 °C for 25 s, annealing at 61 °C for 30 s, and extension at 72 °C for 50 s (modified from Ioos et al. [37]). The final extension was performed at 72 °C for 10 min. For detection of *C. ferruginosum*, PCR thermocycling was modified from Lee et al. [56], consisting of an initial denaturation at 95 °C for 3 min, followed by 40 cycles of denaturation at 95 °C for 30 s, annealing at 70 °C for 30 s, and extension at 72 °C for 30 s. A final extension was performed at 72 °C for 5 min.

Because the plant material was harvested in 2002–2005 prior to our investigation, we amplified four chloroplast specific primers for *P. sylvestris* to confirm that the DNA was of sufficient quality. To achieve this, cpDNA loci of PCP26106, Pt30204, PCP87314, and Pt71936 were analyzed according to Nowakowska [57].

Each PCR product was separated and visualized by electrophoresis using a 1.5% agarose gel stained with Nancy-520 DNA Gel Stain (Sigma-Aldrich, Milwaukee, WI, USA).

The gels were photographed using a transilluminator BIORAD Molecular Imager Gel DocXR+ (Hercules, CA, USA).

### 2.5. Sequencing of the PCR Amplicons

The PCR products were purified using the Clean-up™ kit (A & A Biotechnology, Gdańsk, Poland), according to the manufacturer's instructions. Sequencing was performed using the BigDye Terminal Cycle Sequencing Kit (AB Applied Biosystems, Waltham, CA, USA) in an ABI 3500 Genetic Analyzer (Life Technologies™, Waltham, MA, USA). Species assessment of obtained sequences was performed by comparison to NCBI using BLAST. The criteria used for the identification were as follows: sequence coverage >80%; similarity to taxon level 98–100%; and similarity to genus level 92–97% [4]; and the top matching species was usually chosen for identification.

## 3. Results

Our preliminary study showed that needle samples collected in central and northeastern Poland from dying *P. sylvestris* (sample 1 from private garden, Figure S1) and *Pinus nigra* (sample 8 from ornamental nursery, Figure S1) showed the presence of *D. septosporum* DNA in infected tissues. Ninety-nine percent of sequences from the PCR amplicon of sample 1 (*P. sylvestris*, Table 2, Figure S1A) were identical to *D. septosporum* (GenBank MK993549). The β-*tub2* sequence from sample 8 (*P. nigra*, Table 2, Figure S1B) was 98% identical to *D. septosporum* (GenBank MK993549). Interestingly, the amplicon of EF1-α gene in sample 8 from *P. nigra* (Figure S1B) exhibited 99% sequence similarity to *D. pini* (GenBank KY857608, Table 2). To the best of our knowledge, this is the first report of the occurrence of this species in Poland. In contrast, *L. acicola* and *C. ferruginosum* were not found in any of the nine investigated samples (Table 2, Figure S1C,D).

**Table 2.** Initial testing of pine needles for pathogens using species-specific primers, with positive or negative results of this assessment, and similarity level (%) to the GenBank data.

| Sample * | Host Species | Age of Tree | *Dothistroma septosporum* | *D. pini* | *Lecanosticta Acicola* | *Cenangium ferruginosum* |
|---|---|---|---|---|---|---|
| | | | β-tubulin | EF1-α | EF1-α | ITS |
| 1 | *P. sylvestris* | ~20 | +/99% | - | - | - |
| 2 | *P. sylvestris* | ~80 | - | - | - | - |
| 3 | *P. nigra* | ~20 | - | - | - | - |
| 4 | *P. sylvestris* | ~80 | - | - | - | - |
| 5 | *P. nigra* | ~25 | - | - | - | - |
| 6 | *P. nigra* | ~25 | - | - | - | - |
| 7 | *P. sylvestris* | ~80 | - | - | - | - |
| 8 | *P. nigra* | ~25 | +/98% | +/99% | - | - |
| 9 | *P. sylvestris* | ~80 | - | - | - | - |

* number of samples corresponds to data from Figure S1.

Because the preliminary study on *P. sylvestris* showed only the presence of *D. septosporum* (among four investigated pathogen species), in the main study we only used the β-*tub2* gene marker to assess the presence of this fungus species in needles of the 72 seed trees from nine FDs. Scots pine genomic DNA isolated from long-term stored needles was of acceptable quality (i.e., without inhibitors), because the amplification of four cpDNA loci (PCP26106, Pt30204, PCP87314, and Pt71936) resulted in high yields of target-sized PCR fragments (data not shown).

The *P. sylvestris* stands in which *D. septosporum* has been found to date are mainly located in southern Poland (Figure 2). Nevertheless, our research conducted in seed stands did not confirm the infestation of seed trees, which are thus far free from the pathogen under study. Furthermore, the investigated marker of *D. septosporum* (Table 1) was negative in all asymptomatic needles from *P. sylvestris* seed trees (Figure S2), which suggests that

the pathogen does not lead a latent life as an endophyte, or simply was not present in the samples. This last research is especially important to foresters who use selected spruces as seed trees.

## 4. Discussion

### 4.1. Emerging Pine Needle Pathogens

Preliminary work in our study attempted the early detection of four fungal pathogens (*D. septosporum*, *D. pini*, *L. acicola*, and *C. ferruginosum*) in two pine species, *P. sylvestris* and *P. nigra*, to select needle pathogens to be tested in the main study. Because *L. acicola* and *C. ferruginosum* were not found in our investigation, it was natural to focus on the genus *Dothistroma* in the main study. *Dothistroma pini* had not been found in Poland until this study on black pine in an ornamental nursery. Thus, further studies were conducted to search only for *D. septosporum* in *P. sylvestris* seed trees. Another reason why we did not assess other fungal species in the needles of pine seed trees was that, at the time of our study, we found no reports of *D. pini* or *L. acicola* in the literature in Poland, and the last report of *C. ferruginosum* in Poland dated back to the 1980s [49]. Therefore, we assumed that it was most important to evaluate the presence of *D. septosporum*, and to confirm this presence by molecular methods.

The lack of *D. septosporum* in all tested *P. sylvestris* trees (main study), allowed us to achieve our objectives. Thus, we were able to reject the hypothesis that needles from "mother trees" used for seed collection and subsequent nursery raising of seedlings may be infected with this new invasive pathogen, *D. septosporum*, which has been found in other pine forests in southern Poland [35]. The selection of pine seed stands was based on genetic studies conducted using microsatellite nuclear and mitochondrial markers, which yielded an average heterozygosity level of $H = 0.815$ and a diversity coefficient $F_{ST(WC)} = 0.026$ [40], representative of other forests in Poland, so that the seed source matched planting locations.

### 4.2. Accuracy of the Molecular Methods in D. septosporum Detection

The need to use molecular methods for detection or identification of plant pathogens arises from the frequent absence of disease symptoms, e.g., at early stages or in asymptomatic infections, and the paucity of morphological differences among related pathogens. However, by using markers specific for these fungal species, it is possible to detect the fungus even in the absence of visible disease symptoms, which is the case in early stages or in climatic conditions unsuitable for the pathogen, or if the pathogen shows latent asymptomatic infection. *Dothistroma septosporum* can only be distinguished from *D. pini* by molecular techniques [42,58]. When comparing molecular methods for the detection of *D. septosporum*, previous research [59] has shown that real-time PCR is more accurate than conventional PCR for this type of study. Real-time PCR showed lower contamination levels than conventional PCR, which in turn has higher sensitivity and versatility. In our study, the precision of PCR detection with species-specific primers, and consideration of the cost of the method while desiring to test 72 individuals (with two replicates per seed tree), were key factors in the decision to use the conventional PCR method. Based on the work of Fabre et al. [17], it would have been possible to test a significantly smaller number of samples, because testing each sample would have led to confirmation of the consistency of the results obtained (which were negative in this case), but this would have been considered less efficient for replicates. The fact that we were able to detect *D. pini* should be considered an important finding, even if only in one case, i.e., on *P. nigra* from the ornamental nursery in Osuchów. To our knowledge, this is the first confirmation of the occurrence of *D. pini* in Poland. Transmission of pathogens from ornamental nurseries to forest nurseries is a characteristic route for the introduction into forest stands [60–62]. For this reason, more comprehensive monitoring should be conducted to confirm the presence of this species on black pines, and perhaps also to identify the pathogen on Scots pines.

### 4.3. Pathway of Spread of D. septosporum in Europe

We examined seed infection and were interested in this potential means of spread for the pathogen. However, other routes of transmission are known, e.g., through seedlings and soil or through wind and rain, or machinery, vehicles and shoes [63]. Some plant endophytes are known to be transferred by a process known as vertical transfer under the seed coat [42]. In contrast to horizontal transfer, i.e., direct transfer of an endophyte from parent to offspring, which can promote cryptogenic behavior towards pathogenicity, transfer under the seed coat provides a ready source of inoculum for germinating seeds [43,44]. Although it has not been confirmed that pathogens such as *D. septosporum* can be transmitted under the seed coat, it is known that spores of pathogenic fungi can be transferred to pollen [45,46]. The presence of endophytic fungi depends on the age of the shoots, their position in the crown, and the type of tissue [64]. Investigations by Kowalski and Zych [41] revealed the qualitative and quantitative composition of mycobiota as a function of tissue type, age of asymptomatic shoots, and their location in *P. nigra* crowns. Some fungi, such as *Cyclaneusma minus* (Butin) DiCosmo, Peredo & Minter, a pathogen of needles, are able to infect and behave as an endophyte throughout their life cycle as found in *P. radiata* [47]. Furthermore, latent infections in *P. nigra* and *P. sylvestris* by *D. sapinea* without symptom development have also been observed [48]. In addition, tolerance can describe a situation in which symptoms develop but growth remains unchanged [65], and a loss of some foliage is observed associated with only a slight effect on photosynthetic capacity and growth. In both cases, the host plant benefits from the presence of the latent pathogen, which inhibits potential competitors. To date, no detailed studies have been published on the possibility of *Dothistroma* as an endophyte, so we attempted to examine this possibility with our sampling of seed trees. In our case, we did not find evidence of endophytism by *Dothistroma* in needles, but further study with a larger number of samples is needed. Although we did not test the seeds themselves in our study, we believe that the mentioned risk is low because we did not find any endophytic behavior of the fungus in the needles.

### 4.4. European Approach Problem Solving

The ability of *D. septosporum* to infect various coniferous species poses a genuine threat to forests globally, and in Poland, and particularly black pine stands. Similarly, the accidental introduction of *D. pini*, which is clearly a serious pathogen of *P. nigra*, may have serious negative consequences for European forests. The introduction of *D. pini* into temperate areas where it has not yet been recorded also poses a threat to *Pinus*. *Dothistroma septosporum*, recorded in the EU phytosanitary categorization as *Scirrhia pini* [11], became so common in Europe that it was eventually removed from the quarantine list of the European Union [63]. According to this provision, a pine species highly susceptible to *D. septosporum*, such as *P. nigra* [63], cannot be used for afforestation purposes in Poland, thus reducing the risk of spread of the pathogen among *P. sylvestris* in southern Poland. EFSA concludes that the introduction and spread of *D. septosporum* and *D. pini* is not fully prevented by Council Directive 2000/29/EC, despite the general ban on the import of host plants of *D. septosporum* and *D. pini* from non-European countries, and the restrictions on the import and trade of *Pinus* plants intended for planting specific to *D. septosporum* [63]. In particular, the possibility of importing host plant material other than *Pinus* plants for planting from non-EU European countries, and to move such material within the EU territory, allows the introduction and spread of these pathogens to parts of the EU where they are currently not known to occur. If all rules were removed, the level of imports of infected host plants (whether intended for planting or not) into the EU would increase, leading to an increased rate of introduction and spread of *D. septosporum* and *D. pini* into the risk assessment area [63].

Decision 1104/2012/EU of the European Parliament and Council of 21 November 2012 clarified that Forest Reproductive Material (FRM) must be free from harmful organisms (but without specifying the pathogen species). The Law on Forest Reproductive Material

of 07.06.2001 (Act, 2001) regulates the marketing of FRM by introducing restrictions, prohibitions, and controls with fines. Forest reproductive material listed as "commercial seed" and "selected seed" in the registers of the State Forests National Forest Management Company and National Parks must be registered [51,52]. These regulations tighten control of seed destined for plantations by establishing rules for its use within and outside state borders. In addition, this law introduced regionalization of seed production, which also applies to *P. sylvestris*. All the pine trees assessed in this study originated from select stands, which according to the EU Directive (1999/105/EC) are "trees used for the production of progeny by controlled or free pollination of one of the identified parent individuals used as females with pollen from one parent individual or more than one identified or unidentified parent individual".

### 4.5. Influence of Climatic Changes on Dothistroma Distribution

Recently, an increase in the global mean annual temperature has been observed, e.g., 2020 was the second hottest year on record, and all five of the hottest years on record have occurred since 2015 [66]. In Poland, we have also observed climatic changes manifested by an increase in annual mean temperature during the growing season, and periodic drastic droughts in 2015. This may have a similar impact on the occurrence of the forest pathogens to that of bark beetles, such as *Ips typographus* in the Białowieża Forest [67]. As a result of changing global climatic conditions, an adaptation of pathogens to the Northern Hemisphere has been increasingly observed for pathogens that normally attack coniferous species from warmer climates. Another ascomycete fungus, *Fusarium circinatum*, which was known to preferentially infect *Pinus* spp. species from the Mediterranean region (Spain, Italy, and southern France), is now commonly found on pine seedlings in central and northern Europe [4]. The influence of climate on disease development can indeed mask the presence of *Dothistroma* spp. in regions that are marginal and not conducive for disease development [68]. These countries have experienced warmer and wetter climates over the past two decades, which may have contributed to the increased prevalence of disease. Increased sampling and efforts by researchers to search for DNB have led to recent discoveries of the pathogen in north-central Europe [29,35]. This indicates that it is important to routinely screen plant material for invasive pathogens, even if disease symptoms are not visible.

## 5. Summary and Conclusions

The currently observed climatic warming both weakens the hosts and favors the development of pathogens, thus forcing the development of new, effective methods for their detection in the environment. There are several methods of monitoring the pathogens infecting pine species. Among these, the most efficient and accurate are tests based on molecular PCR detection, independently of the presence or absence of disease symptoms. Among the four pathogenic fungi investigated, i.e., *D. septosporum*, *D. pini*, *L. acicola*, and *C. ferruginosum* in private gardens (Konstancin Jeziorna) and an ornamental nursery (Osuchów) in Central Poland, *D. septosporum* was detected on *P. sylvestris*, whereas *D. septosporum* and *D. pini* were discovered on *P. nigra*. For the first time, *Dothistroma pini* was found in Poland, which could raise concerns and requires further extensive investigation in coniferous stands. Based on this preliminary finding, our further research focused on *P. sylvestris* seed stands from southern Poland, where the occurrence of DNB has been reported in commercial forests. The assessed seed trees, considered to be a source of seeds and FRM in forestry, were free of the tested pathogens screened with specific markers. The lack of *D. septosporum* in all tested needles demonstrates that the pathogen did not show an endophytic behavior, suggesting that the assessed seed trees could be categorized as "free of pathogens". This is consistent with the condition stipulated for FRM included in the Polish and EU guidelines. Our results confirm the importance of conducting molecular tests to detect *D. septosporum* in needles of seed trees, which would facilitate a breeding

program for *P. sylvestris*, particularly in the changing climatic conditions that favor the global spread of *Dothistroma* spp.

The introduction of new species into plantings or forest stands, e.g., those least susceptible to *Dothistroma* spp. or clones resistant to this pathogen, will be an important step in future management of DNP [69]. An important and potentially highly effective option for the future sustainable management of DNB and related diseases is prudent selection of planting material, although resistance (tolerance) is not the only factor to consider. The study of genetic structure and diversity in pathogen populations can aid in the development of effective disease control and management strategies. A pathogen with high genetic variability has strong evolutionary potential and is likely to be able to rapidly adapt to new conditions, such as environmental changes or changes in host resistance [70].

**Supplementary Materials:** The following are available online at https://www.mdpi.com/article/10.3390/f12101323/s1. Table S1. Locations and site types of pine stands in which plant material was collected for *D. septosporum* DNA analyses, Figure S1. Preliminary results of a search for pathogen presence as a cause of pine needle dieback in gardens, nurseries, and forests. A: fragment showing presence of *D. septosporum* in *P. sylvestris* (231 bp, sample 1, weak signal), and *D. septosporum* presence in *P. nigra* (231 bp, sample 8). B: presence of *D. pini* in *P. nigra* (193 bp, sample 8). C: Attempts to detect *L. acicola* with negative results. D: Attempts to detect *C. ferruginosum*. L-100 bp DNA Ladder (Sigma-Aldrich, Milwaukee, WI, USA). Arrows indicate fragments of 231 base-pair length from different DNA extract samples numbers 1 to 9, Figure S2. Assay for *Dothistroma* PCR products from needles of Scots pine seed trees. Lane P (A, B, and C): positive control ca. 231 bp). Other lines (1–72) are PCR of DNA extracts from different seed trees showing absence of amplicons; L-FastRuler™ Middle Range DNA Ladder (Fermentas, Waltham, MA, USA).

**Author Contributions:** Conceptualization, T.O. and J.N.; methodology, J.N., T.M., L.B. and T.O.; validation, T.O., S.B., T.M., T.H., E.P.-N. and J.N.; formal analysis, P.W., J.N. and T.M.; resources, J.N., T.M. and T.O.; data curation, T.O., S.B., E.P.-N. and T.H.; writing—original draft preparation, P.W.; writing—review and editing, T.O., T.M., J.N., L.B., T.H., S.B. and E.P.-N.; visualization, P.W., T.M. and J.N.; supervision, J.N.; project administration, T.O.; funding acquisition, T.O. and S.B. All authors have read and agreed to the published version of the manuscript.

**Funding:** The study was supported by Forest Research Institute (IBL) in Sękocin Stary (Poland).

**Institutional Review Board Statement:** Not applicable.

**Informed Consent Statement:** Not applicable.

**Data Availability Statement:** Not applicable.

**Acknowledgments:** Pola Wartalska is grateful for the IBL authority which allowed her 3 week stay in the Department of Forest Protection, IBL in Sękocin Stary where some of the research was conducted. The authors thank Marek Siewniak for providing pine needle samples for preliminary analyses, and to Jan Kowalczyk and Jan Matras (Department of Silviculture, IBL, Poland) for the selection of seed trees in the investigated area. Marek Lisańczuk provided the geodata information.

**Conflicts of Interest:** The author Pola Wartalska is an employee of MDPI; however, she did not work for the journal *Forests* at the time of submission and publication. The funders had no role in the study's design, in the collection, analyses, or interpretation of data, in the writing of the manuscript, or in the decision to publish the results.

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
