# Peer review of "Dothistroma septosporum Not Detected in Pinus sylvestris Seed Trees from Investigated Stands in Southern Poland"

_forests, doi:10.3390/f12101323_

Round 1
Reviewer 1 Report
I believe that this research is very important for the conservation of ecosystems in Europe.
It would be better if you add the following information.
- In the introduction, it would be good to mention other examples where exotic fungi have had a negative impact on ecosystems.
- In Table 2, β-tubulin is generally considered to be a gene with a very slow evolutionary rate, so the value should be as high as 99 %, but can 98 % be considered high? Also, how much variation is known in this gene at the sequence level?
- In section 4.3, as for how D. septosporum spread in Europe, you cite previous studies that it spread via seeds, but are there any other possible expansion routes? For example, is it possible that the fungus spread via the soil or the atmosphere?
- In section 4.4, how are the boundaries of the restrictions on tree transfer determined? Is it by experience, by law, or by research on genetic diversity? It will be more convincing if you present the literature information on the basis of this.
Author Response
Reviewer #1
COMMENT 1. In the introduction, it would be good to mention other examples where exotic fungi have had a negative impact on ecosystems.
RESPONSE: Such information is now provided in the Introduction and several new paragraphs have been placed in the Introduction. “Pine is an important component of native forests and plantations in Europe, where it has both economic importance and an important ecological role. Alien invasive pathogens of conifers such Lecanosticta acicola, Sphaeropsis sapinea or Fusarium circinatum have recently caused the death of many pine trees in Europe [1, 2]. Some oomycetes, e.g. Phytophthora cinnamomi, or fungi, e.g. F. circinatum, now found only in southern Europe (mainly in Spain and Portugal), can easily colonise new geographical locations under the observed global warming, posing the risk of further northward spread [3]. Some studies have shown that this pathogen could also cause a lot of damage in Poland if it is introduced there [4].”
COMMENT 2. In Table 2, β-tubulin is generally considered to be a gene with a very slow evolutionary rate, so the value should be as high as 99 %, but can 98 % be considered high? Also, how much variation is known in this gene at the sequence level?
RESPONSE: Although beta tubulin is a highly conserved gene, BLAST search of sequences similar to different D. septosporum strains showed that its identity could be as low as 96.16%, e.g. strain CMW44656 (MK015049.1) vs. CMW 10930 (AY808196.1). The other species, such as Cercospora kikuchii, C. beticola, C. sojina, and Mycosphaerella coffeicola, share even lower sequence identity, i.e. between 93.78% - 92.03%,to beta-tubulin gene sequence of D. septosporum.
COMMENT 3. In section 4.3, as for how D. septosporum spread in Europe, you cite previous studies that it spread via seeds, but are there any other possible expansion routes? For example, is it possible that the fungus spread via the soil or the atmosphere?
RESPONSE: Other ways of spread has been discussed in section 4.3. as follows “We studied the seeds, so we were interested in this way of spreading the pathogen. However, other routes of transmission are known, e.g. through seedlings and soil or through wind, rain or water used to water plants in nurseries (from rivers in the vicinity), machinery, vehicles, shoes, etc. [EFSA 2013].”
COMMENT 4. In section 4.4, how are the boundaries of the restrictions on tree transfer determined? Is it by experience, by law, or by research on genetic diversity? It will be more convincing if you present the literature information on the basis of this.
RESPONSE: In this section, new information was added as follows “EFSA concludes that the introduction and spread of D. septosporum and D. pini is not fully prevented by Council Directive 2000/29/ EC despite the general ban on the import of host plants of D. septosporum and D. pini from non-European countries and the restrictions on the import and trade of Pinus plants intended for planting specific to D. septosporum [EFSA 2013]. In particular, the possibility to import host plant material other than Pinus plants for planting from non-EU European countries and to move such material within the EU territory allows the introduction and spread of these pathogens to parts of the EU where they are currently not known to occur. If all rules were removed, the level of imports of infected host plants (whether intended for planting or not) into the EU would increase, leading to an increased rate of introduction and spread of D. septosporum and D. pini into the risk assessment area [EFSA 2013].”

Reviewer 2 Report
The paper is related to the problem important from the practice of the pine forest management and conservation - epidemiology of the dothistoma needle Blight (DNB). The research was divided on the two stages: preliminary assesment and the main study - such approach is really worth to be appreciated. Methodes of analyses are rather cleary described, and majority of results properly presented.
The paper, however, contains some ambiguities that should be clarified before it is accepted for publication:
- Criteria for choosing the study places, both - in preliminary assesment as well as in main study - should be more cleary described. The figure 2 is little bit unclear.
- Tittle is a little bit confusing. It is related to the southern Poland, whereas study point were loceated also in the norten part of the country. Moreover, according to the Figure 2. at some southern study points (Kamyk, Czernichów, Domiarki and NiepoÅ‚omice) D. septosporum had been detected.
The Discussion is to broad and and little bit repetitive. The section 4.1 and 4.3 should be conbined it one and some general information from them should be moved to the inroduction.
Author Response
Reviewer #2
COMMENT: Criteria for choosing the study places, both - in preliminary assesment as well as in main study - should be more cleary described. The figure 2 is little bit unclear
RESPONSE: The Section 2 Materials and Method, was revised with additional information about criteria of sample selection. "The selection criteria for the preliminary study was broad because there was limited information about potential DNB." (revised in sec 2.1). Details about the main study are presented in section 2.2 as follows: " the selection criteria for the main study were carefully chosen based on the available literature [Piotr Boron et al. 2016] and the information received during the international COST FP1102 Action - Determining Invasiveness And Risk Of Dothistroma (DIAROD). They showed that D. septosporum is present in commercial stands of both Scots pine and Black pine in southern Poland. For this reason, trees were sampled in stands in the vicinity of commercial stands where disease has been detected.
COMMENT: Tittle is a little bit confusing. It is related to the southern Poland, whereas study point were loceated also in the norten part of the country
RESPONSE: The title has been adjusted to “Dothistroma septosporum not detected in Pinus sylvestris seed trees from investigated stands in Southern Poland”, which better corresponds to the article content. Only preliminary studies included samples from the central and northeastern part of Poland, and the main study focused on pine stands in southern Poland.
COMMENT: Moreover, according to the Figure 2. at some southern study points (Kamyk, Czernichów, Domiarki and NiepoÅ‚omice) D. septosporum had been detected
RESPONSE: That is correct, but the detection of D. septosporum in southern Poland (Kamyk, Czernichów, Domiarki and NiepoÅ‚omice) was done by Piotr BoroÅ„, and motivated us to undertake research in the neighbouring selected seed stands (WDN). Investigated stands were selected in southern Poland not adjacent to commercial stands (Figure 2) where disease had been detected, but where seeds trees were available for sampling.
COMMENT: The Discussion is to broad and little bit repetitive. The section 4.1 and 4.3 should be conbined it one and some general information from them should be moved to the inroduction.
RESPONSE: The Discussion part has been greatly reduced and reorganised as suggested.

Round 2
Reviewer 2 Report
I'm really convinced, that criteria of the study plots should be described in more details. In case of preliminary assessment authors wrote:
"(...)
he aim of this preliminary study was to find out whether target fungi (nedlee pathogens) were abundandt in the plant material. The selection criteria for the preliminary study was broad because there was limited information about potential DBN. In the course of our consultation assesment at the request of both private and public entities, some needels were collected from the dying Scot pines (...)"
Then the sampling points are listed.
However, after such a description, does not explain some important points: What entilies were asked to collect samples? Why specificially they were chosen? Based on the literature data? Because of the information included in any forest related databases? Any another reasons?
Simillary in the description of the mains study.
"(...)
The selection criteria for the main study were carefully chosen based on the aviailable literature [35] and the information we received from the international COST-Action FP1102.
(...)"
I really do believe, that authors chosed criteria carefully, however i would expect more precisous information.
Author Response
We thank the anonymous reviewer for the comments, and here present our answers to them:
COMMENT: I'm really convinced, that criteria of the study plots should be described in more details. In case of preliminary assessment authors wrote:
"(...)
he aim of this preliminary study was to find out whether target fungi (nedlee pathogens) were abundandt in the plant material. The selection criteria for the preliminary study was broad because there was limited information about potential DBN. In the course of our consultation assesment at the request of both private and public entities, some needels were collected from the dying Scot pines (...)"
Then the sampling points are listed.
However, after such a description, does not explain some important points: What entilies were asked to collect samples? Why specificially they were chosen? Based on the literature data? Because of the information included in any forest related databases? Any another reasons?
RESPONSE: "The criteria for the preliminary study arose from information received by the Polish IBL Forest Protection department on the disease symptoms of pine trees in different regions of the country. Therefore, informants were asked to send samples of symptomatic pine trees from which needles were collected for further laboratory analysis." This information has been added into the revised manuscript in section 2.1 Preliminary Assessment.
COMMENT: Simillary in the description of the mains study.
"(...)
The selection criteria for the main study were carefully chosen based on the aviailable literature [35] and the information we received from the international COST-Action FP1102.
(...)"
I really do believe, that authors chosed criteria carefully, however i would expect more precisous information.
RESPONSE: The selection criteria were based on generally accepted practices for sampling of plant material for genetic studies. Such information has been was added in Material & Methods, section 2.2.1. “The plant material for DNA testing (needles) was obtained from the selected trees (mother trees) randomly distributed in a given stand, at least 25 m away from each other, to avoid close kinship between the studied individuals, while sharing good health conditions. These seed trees had been identified by the Polish State Forest Administration as possessing exceptional characteristics”.
This following contains more detail that we have not added into the manuscript. All investigated trees came from “selected seed stands” (WDN in Polish) which were identified as such by the State Forestry Administration for their exceptional characteristics, and which are worth retaining in further forest management. Selected trees from which cones are collected for seed have their own identification passport and even genetic sequences are available. They are characterised by exceptional growth, branch-free, slender trunks and are selected only by a committee of authorised individuals (the same specialists for the entire country). These selected trees (mother trees) provide new propagation material which should be free from harmful organisms according to the directive EC.
Each tree species occurs in Poland in areas of origin, from which the seeds are obtained and where the species occurs naturally. There are selected conservation areas (WDN) of pine, from which reproductive material is obtained following the framework of the "Programme for the Protection of Forest Genetic Resources and Selective Breeding of Forest Trees in Poland for 2011-2035" (Matras et al. 2003, Matras 2005). These guidelines have been implemented in State Forests, and form the basis for maintaining high genetic diversity in restored stands and protection against an influx of forest reproductive material of foreign origin.
As mentioned in the manuscript, some investigated pine stands in southern Poland were selected for baseline research because information was available that pines in economically significant stands were being damaged by the fungus Dothistroma septosporum. This information was provided in the framework of the COST DIAROD action and published by Piotr Boroń.
